# Experiences of Self-Stigma in People with Chronic Psychosis: A Qualitative Study

**DOI:** 10.3390/ijerph20095688

**Published:** 2023-04-29

**Authors:** Tatiana Arboleya-Faedo, Ana González-Menéndez, David González-Pando, Mercedes Paino, Fernando Alonso-Pérez

**Affiliations:** 1ISPA—Health Research Institute of the Principality of Asturias, 33011 Oviedo, Spain; 2Department of Psychology, University of Oviedo, 33003 Oviedo, Spain; 3Faculty of Nursing, University of Oviedo, 33394 Gijón, Spain

**Keywords:** mental health nursing, psychotic disorders, qualitative research, social stigma, self-stigma, looping effect

## Abstract

We present the results of a phenomenological study understanding the personal meaning of self-stigma in people with chronic psychosis. Self-stigma is a frequent phenomenon in the lives of people with psychosis and their families and it functions as a barrier to recovery. Semi-structured in-depth interviews were conducted with fourteen outpatients that suffer from chronic psychosis during January 2020. Data analysis was carried out using an inductive approach as described by Graneheim and Lundman through the MAXQDA 2022 program. The themes observed were: “Contextual Stigma”, “Components of Self-Stigma”, “Skills Loss” and “Coping with Self-Stigma”. The main categories and subcategories were avoidance and escape behaviours from their social environment, labelling, loss of social relationships, negative impact and self-concealment of the diagnosis. Our results revealed influence on each other, forming a looping effect that explains and amplifies the lived experience of self-stigma. These findings highlight the need to implement strategies in nursing practice aimed at training the acceptance and distancing necessary to minimize the impact of self-stigma on people with chronic psychosis. This study adheres to the EQUATOR guidelines for the Consolidated Criteria for Reporting Qualitative Research (COREQ).

## 1. Introduction

Schizophrenia is a challenging subject for human understanding and scientific explanation [1]. The etiological hypotheses of the disorder are many and varied and point in general to the interactive effect of genetic, biological, psychological and social factors [2]. Schizophrenia is characterized by abnormalities in the perception of thinking, emotions, language and behaviour [3]. Schizophrenia accounts for only 30% of the outcome fraction of a much broader spectrum of psychosis [4]. Psychosis is a common and functionally disturbing symptom of many psychiatric conditions [5]. The concept of psychosis has led to the transformation of a medical term to a loaded layman’s term with various negative connotations [4].

Social stigma in people with psychosis is a phenomenon of great interest because of its negative consequences in several domains of social functioning, quality of life and received and perceived social support [6]. Social stigma affects symptom management, disease awareness and adherence to treatment [7]. It is often described as prejudice and discrimination caused by negative stereotypes associated with the diagnostic label such as dangerousness, unpredictability and incompetence [8,9]. People with psychosis tend to endorse these stereotypes, anticipate social rejection and incorporate these views into their own value system, known as self-stigma (SS) or internalized stigma [10]. SS is therefore a frequent phenomenon in the lives of people with psychosis and their families and often works as a barrier to recovery and the achievement of life goals [11]. The identification of variables predicting SS has been investigated in numerous empirical studies [7,12,13,14]. Two recent meta-analyses [15,16] revealed a significant association between SS and clinical and psychosocial variables: psychotic and depressive symptomatology, subjective recovery, functioning and adherence to treatment, hope, self-esteem, empowerment, self-efficacy, quality of life, and social support and integration.

Despite its interest, qualitative studies on the effects of SS in people with psychosis are still scarce [17,18]. Most of these studies report that the SS of people with psychosis emerges as a consequence of labelling, fear of mental illness in society and public discrimination [17,18]. Diagnosed people respond emotionally to social stigma with feelings of worthlessness and inferiority, and consequently impose social distancing, isolation and alienation behaviours on themselves [19,20]. Studies also agree on the loss of social roles, lack of opportunities and concealment of the diagnosis to avoid stigma [19].

From this perspective, it is important to describe and understand the phenomenon of SS. We propose this research question: How do people with chronic psychosis live SS experiences? In this case, the phenomenological method is particularly appropriate when trying to understand the personal meaning of the lived experience. Phenomenology seeks the essence of lived experience, without prejudice, looking for the way to understand how the person has lived it, reflects on it and gives meaning to that lived experience. The context that forms the lived experience consists of time, space, person who lives that experience and interaction with others and the world that surrounds him. People are considered linked to their world (incarnated) and are only understandable in their context [20]. According to Husserl [21], a phenomenological approach consists of decomposing the parts of inner perception in order to describe it properly. Furthermore, this approach is consistent with the holistic and humanistic view of the nursing performance, encouraging the understanding of the SS process and can help in the planning and optimising the necessary support and care [22].

Therefore, this study aims to explore the SS-related experiences of people with chronic psychosis.

## 2. Materials and Methods

### 2.1. Design

We conducted a qualitative study from a phenomenological approach [21]. A descriptive phenomenological analysis was applied; the aim of this analysis was to describe experiences and leave aside or ‘bracket’ the researcher’s perceptions in order to enter the life world of the research participant without presuppositions [23]. 

The present report adheres to the EQUATOR guidelines for reporting research, following the Consolidated Criteria for Reporting Qualitative Research (COREQ), a 32-item checklist for interviews and focus groups [24]. 

### 2.2. Sampling and Recruitment

The research was conducted with a sample of outpatients from a Mental Health Public Service in Asturias (Spain) that provides mental health care to a population of 50,000 people in the context of a city of 250,000 inhabitants. The sample was based on a previous study [25] in which 103 patients were assessed using the Internalized Stigma of Mental Illness Scale (ISMI) [26]. To be included in this study, participants were required to comply a number of inclusion criteria such as: being over 18 years of age; being able to express themselves verbally; meeting diagnostic criteria for the International Classification of Diseases Tenth Edition (ICD-10) codes F20–F29 for schizophrenia, schizotypal disorders and delusional disorders [27] for at least two years and presenting with elevated levels of SS (scores ≥2.5 in subscales of alienation and approval of stereotypes of ISMI). Measures of Perceived Discrimination, Social Withdrawal and Stigma Resistance were omitted as they do not constitute dimensions of internalized stigma per se. The exclusion criteria were signs of severe physical illness, acute psychosis episode, cognitive impairment, intellectual disability and/or primary diagnosis of substance dependence.

The participants were selected using theoretical and purposive sampling. We recruited participants of different socio-demographic characteristics in order to obtain as diverse set of narratives as possible. Since the power of the sample does not lie in its size, but rather in the ability to reflect the diversity of the phenomenon studied, the sample size was not determined beforehand, but rather during the course of the study. People were contacted by telephone to ask if they wanted to participate. Through face-to-face interviews, all participants received information about the study with emphasis on its voluntary nature and were offered the opportunity to ask questions and make clarifications before signing the informed consent form. The participants had the possibility to choose a time and place to promote a climate of trust. No incentives were offered for participation. All were guaranteed confidentiality of data and the possibility to withdraw from participation at any time during the interview. Data saturation was reached with the participation of 14 people when we detected that no new relevant information was emerging [28]. The socio-demographic and clinical characteristics of the participants are shown in Table 1 and Table 2, respectively.

### 2.3. Data Collection

Data collection was conducted through semi-structured in-depth interviews, which is the main method of data collection in phenomenological methodology [28]. Sociodemographic and clinical data were recorded for all participants. The participants had knowledge about the reasons for participating in the research. The researchers developed a script of guiding questions (Table 3). These questions were not themes or characteristics of the phenomenon. The interviewer was able to ask new questions and explore the questions in more detail, according to the speech of each participant. The interviewers used open questions as much as possible to allow for the participants to express their experiences openly, favouring the understanding of their subjectivity and uniqueness. The interview began by justifying the project and inviting the participant to a deep reflection on their personal experiences, guaranteeing confidentiality, respect and anonymity. It began with a general opening question so as not to condition the participant: What is your day-to-day like? The interviews were carried out in the Mental Health Center where the patients were receiving treatment, in a quiet room different from the usual consultation, during the month of January 2020. Fourteen interviews of sixty minutes’ duration were conducted under the guidance of a nurse specialist in mental health. All interviews were audio recorded and subsequently transcribed. Findings presented in verbatim quotes are followed by the participant’s identification code after recoding to ensure anonymity. 

### 2.4. Data Analysis

Data analysis was carried out concurrently with data collection using qualitative content analysis with an inductive approach as described by Graneheim and Lundman [29]. The unit of analysis was the interview. The analysis started with reading the data in their entirety several times and continued with dividing the data into meaning units, consisting of words or sentences related through content and context. Each meaningful unit was summarized into a condensed meaningful unit, and the initial codes appeared (Table 4). A total of 200 initial codes were extracted from the interviews. The initial codes were classified as subcategories based on the similarity of the concept. Subcategories were classified into a category. The category refers mainly to a descriptive level of content and can thus be seen as an expression of the manifest content of the text. Lastly, the categories were organised into themes. A theme can be seen as an expression of the latent content of the text. The transcripts were individually coded by the same member of the interview team who conducted the interviews, with a subsequent review by another researcher. The coding system was constantly reviewed during the analysis. An external researcher replicated the coding of the transcripts. Discrepancies in the coding and refinement of the categories were discussed by the team and resolved by consensus. The program used for the analysis was MAXQDA 2022 (VERBI Software, 2021; VERBI Software. MAXQDA 2022 [computer software]. Berlin, Germany, 2021) [30].

### 2.5. Rigour

To ensure the rigour of the study, the criteria of credibility, transferability, consistency and confirmability were followed [31]. The interviews were conducted by a nurse specialist in mental health with over 11 years of clinical experience and extensive training in qualitative research. To enhance credibility, the role and possible biases of the researcher were acknowledged. There was a relationship established with participants prior to study. The researcher maintained an open view, leaving prejudices behind, minimising the interference of subjectivity in the research activity. The interviews and analysis were conducted in Spanish. Although the results reported in the article were translated to English, the underlying latent meaning was preserved; the interpretations and translation have thoroughly been reflected on and discussed by the research team. Results were also returned to participants for validation. In terms of transferability, a detailed description of the characteristics of the context and the type of sample was carried out. Saturation of the data was sought. Triangulation of the researchers was used for data analysis, providing different approaches by an interdisciplinary team. An external researcher reviewed the construction of the methodology and data analysis for data consistency. A detailed description of the process of data collection, analysis and interpretation was provided with a constant comparison of the emerging results with previous studies. To achieve confirmability, the study was submitted to the Research Ethics Committee of the Principality of Asturias, describing what the interaction with the study participants would be like. The interviews were recorded and transcribed verbatim by the interviewer herself, using field notes and reflective notes to avoid missing or misinterpreting any data. The results were cross-checked with the existing citation of sources.

## 3. Results

The results revealed 31 subcategories, 15 categories and 4 themes. The themes observed were: contextual stigma, components of self-stigma, skill loss and coping with self-stigma. The results are presented in Table 5.

### 3.1. Theme: Contextual Stigma

This refers to the social contexts that may influence the development of SS. Four categories emerged from this theme: social stigma, social stigma per se, loss of social relationships and loss of social roles.

#### 3.1.1. Social Stigma

All participants reported witnessing stigmatising comments, some consider that others believe that they may be faking the illness. 

“They said that crazy people had to be in the loony bin and I’m just saying I think I’m not crazy, I have a disease” (C0090)

“People associate schizophrenia with being crazy” (C0026)

“They think I’m playing the victim or that I’m a crybaby or they tell me that what you need is to have been brought up differently or if only they give you a few slaps in time or they should have beaten the depression out of you…” (C0104)

Additionally, that it is an untreatable disease and requires hospitalisation. They speak of prejudice where they explain that they feel fear, mistrust and misunderstanding towards them and of discrimination and experiences of rejection.

“They are afraid that we will harm them…” (C0090)

“They don’t trust me” (C0019)

“Taboo subject that you tell people and people don’t understand” (C0026)

“I had a very bad time because I saw that everyone was ignoring me” (C0063)

“When you start to get to know someone, you start to tell them that you were sick or that I’m mentally ill, they start to pull away from you“ (C0045)

They feel they are treated differently, lack of credibility, compassionate treatment and indifference. 

“When you tell them one thing, it’s as if you have no credibility” (C0026)

“The few people who know about it look at me with compassion, as if to say let’s see this…” (C0090)

“They don’t talk to me or anything” (C0028).

They consider that the media are responsible for disseminating a public image of the dangerousness of the mentally ill.

“There are things on television about… a schizophrenic killed so many people and people are afraid of me because they know what I have” (C0028)

“In the news there is always the phrase about suffering from mental disorders” (C0020)

Some of the interviewees refer to family stigma due to the lack of understanding of the illness, some relatives also hide the diagnosis from others. There are even participants who suffered a separation with their partner after the first crisis due to rejection.

“It is difficult to understand something like this if apparently you are normal but inside you are broken” (C0001) 

“My husband doesn’t want to tell people my diagnosis… he has depression…” 

Additionally, they express difficulty in finding work and discrimination as a consequence of negative stereotypes about the illness such as incompetence and inferiority. However, others point out their limitations to develop a job. 

“They label you as disabled and dependent and they treat you differently, they label you as inferior” (C0015)

“As soon as you tell them that I had a mental illness, they don’t accept the job” (C0045)

“I don’t think about it anymore because I have come to the conclusion that I can’t work” (C0019)

They speak of a lack of humanization, barriers and inequality in treatment just for the fact of contextualising a behaviour in a mental health service.

“You are not a number, you are not a folder, you are something more, you are a story” (C0019)

“Why do they have a glass partition, in case they are attacked? Because those below don’t have it…” (C0019)

#### 3.1.2. Social Stigma per se

Some of the participants use social stereotypes towards people with mental illness and show understanding towards the stigmatising people.

“People who kill” (C0020)

“They go crazy, they destroy everything, they destroy a whole room, they destroy a family… I understand people, you have to understand them too, I put myself in their place and I understand “ (C0026)

“I would do the same as people” (C0020)

The influence of family stigma with the social stigma itself is shown.

“I remember when I was a child with my mother crossing the street, you have to cross because they are here, she used to say…” (C0020)

Some avoid relating to them because they remind them of themselves and prevent other people from relating to them.

“People on medication, they are very fucked up people and I don’t want to see that, I want to see good, healthy people” (C0026)

#### 3.1.3. Loss of Social Relationships

We differentiate between relationships with partners, social relationships and family relationships. Most of the participants are single or divorced. They report the difficulty they have in finding a partner because of mistrust, isolation, lack of symptom control and diminished capacities. They also mention the lack of opportunities due to their poor social life, fear of rejection, economic problems and low self-esteem.

“Unthinkable, no, I don’t want to. I trusted my father and my mother, but not others” (C0019)

“I have a tendency to isolate myself, I find it difficult to communicate and I don’t like to socialise and that dynamites everything” (C0001)

“I don’t know how to guide myself, I don’t know how to orient myself or organise things the way other people organise things to go with their partners” (C0104)

“Nobody would want me today” (C0028)

Most of the participants report none or few social relationships related to the loss of friendships due to social rejection when disclosing their diagnosis. They also recognise the difficulty in relating to others and their tendency to self-isolation in relation to psychotic symptomatology. 

“I find it difficult… I prefer to be alone” (C0022)

“And I feel observed today… even today, 24 h a day… because let’s see…, I am at home and where I am best is at home” (C0063)

Most of the participants feel support from their family and have good feelings towards them, although some verbalise incomprehension.

“If it is not professionally, it is very difficult for them to understand you…” (C0001)

#### 3.1.4. Loss of Social Roles

Dropping out of school and loss of work due to illness.

“I stopped studying when I was in my twenties, when the crisis hit me” (C0001)

“With schizophrenia you can’t work because they throw you out of work because you don’t work hard…” (C0026)

### 3.2. Theme: Components of Self-Stigma

It refers to SS as a dynamic process that manifests itself in three dimensions: cognitive, emotional and behavioural. The analysis of this theme revealed five categories: labels, derived emotional responses, behaviour: avoidance and escape, anticipatory discrimination and negative repercussions.

#### 3.2.1. Labels

Participants revealed negative beliefs where the internalization of the stereotype manifests itself in the form of labels such as dangerousness, unpredictability and useless-ness.

“This is the disease of criminals…, I am very aware of it, I have to take it because if I don’t, I might be able to do something terrible, so in order for that not to happen, I have to take the medication and be rigorous. Be careful because in this situation you may become violent, be careful not to do so or you may react in an unpredictable way…” (C0020)

“I feel useless because before I was active, hard-working, I felt strong” (C0045)

#### 3.2.2. Derived Emotional Responses

As an emotional response to labels, alienation appears where participants revealed negative emotions such as shame, feelings of inferiority and disappointment resulting from having the disease.

“I’m ashamed… the fact that they know that I have this illness makes me ashamed…” (C0090)

“I’m very embarrassed when I go to pick up prescriptions at the pharmacy and they see it there… antipsychotics…. “ (C0063)

“I am disappointed with myself” (C0039)

“I think I failed in life” (C0045)

#### 3.2.3. Avoidance and Escape Behaviours

Some interviewees choose social isolation as an avoidance strategy in response to prejudice. 

“I think that I am noticed, that the people who knew me and see me now, know that there is something there, so I isolate myself and that’s it” (C0019)

#### 3.2.4. Anticipatory Discrimination

A large proportion of respondents experienced anticipatory discrimination, believing that others would devalue or reject them for having a mental disorder.

“I’m tired of failures. It would be good for me… I know it would be good for me, but there have been many failures… I am afraid… it took me a while to get used to this life, to being alone… but now I am used to it and I am afraid I am going to fall into the anguish I had when I was alone again…” (C0001)

“I usually say it in case they do not want to approach me” (C0015)

#### 3.2.5. Negative Repercussion

Most participants experienced negative affect and low self-esteem in relation to SS, describing feelings of loneliness, either through loss of social relationships or social isolation; of dependence; of guilt for having the illness; of failure and of lack of confidence. 

“You are so lonely you don’t know who to tell” (C0015)

“I am on my own without depending on anyone impossible” (C0063)

“Sometimes I feel bad for behaving the way I behave with people” (C0014)

“Totally failed because I had one life and now I have another” (C0063)

“My biggest problem is the lack of confidence in myself” (C0039)

Fear also appears in various forms, such as fear of psychopathological decompensation, to partner losses, social rejection and in the context of their own positive symptomatology. 

“Fear of not being understood, of rejection…” (C0090)

“Fear that the medication will fail and that I will get sick inside my head” (C0022)

“I am afraid that people will be afraid of me too” (C0045) 

“I always have those fears there” (C0063).

They also experience a very low self-concept, lack of charisma, sadness and inferiority. 

“I am very weak, I am not able to cope with things” (C0045)

“I feel sorry for myself for having this disease” (C0026)

“No one talks to me…. It’s very sad” (C0028)

### 3.3. Theme 3: Skills Loss

This theme encompasses the loss of cognitive abilities and feelings of incapacity. The analysis shows two categories:

#### 3.3.1. Cognitive Disturbances

They report a loss of conversational ability, lack of concentration, slowness and secondary affective flattening, diminished executive abilities and difficulty in problem solving. 

“I try…, but I don’t engage in conversation” (C0022)

“I can spend a while reading a page and I have nothing left and I start all over again” (C0019)

“With the treatment the emotions practically disappeared” (C0039)

#### 3.3.2. Abandonment of Activities

They also report a tendency to give up activities due to lack of perseverance, tiredness, lack of motivation and low spirits. Participants perceive a feeling of incapacity for activity, apathy and insecurity in carrying them out.

“I don’t feel like doing different things, I don’t have enough vitality” (C0039)

“I don’t feel like doing anything” (C0063)

### 3.4. Theme: Coping with Self-Stigma

This refers to the way of coping with the disease. The analysis shows the following categories:

#### 3.4.1. Passive Coping

One of them is passive confrontation, as in seeking to manage emotional distress; participants speak of resignation and the abandonment of a futile fight against the disease or the search for healing through faith or miracles.

“Good, good, I’m used to it, I’ve been used to it for many years, I’m used to it” (C0001)

“It’s something that happened to me, like someone who has cancer or who is missing an arm, so I’m going to learn to live with it” (C0019)

“I take refuge in some kind of miracle happening, that everything will finally be solved” (C0104) 

#### 3.4.2. Active Coping

Active coping is focused on improving symptom control by reducing drug use, focusing on effort or discipline and following the prescribed treatment and seeking professional help.

“I said to myself, yes, I can go ahead, yes, I can get out of it, I can be a little better” (C0090)

#### 3.4.3. Avoidant Coping

The isolation and self-concealment of the diagnosis, one of the most frequent codes, appears as a consequence of fear of rejection.

“Isolating myself was the best thing to do” (C0019)

“I don’t have any friends… because for once I told a person about it, they turned their back on me, that limits me a lot and I don’t tell anyone…” (C0090)

“I think these things should not be told, that schizophrenia is a taboo” (C0026)

#### 3.4.4. Ineffective Coping

One person reported a self-harm attempt as an ineffective coping response.

“No one wanted me and so I threw myself out of a window…” (C0063)

## 4. Discussion

The main aim of this research was to explore the life experiences related to SS in people with chronic psychosis. This objective, unattainable through any other type of methodology, shows the importance of qualitative research, as it is a way of allowing those truly involved to have their say—in this case, people with chronic psychosis who experience the devastating phenomenon of SS. The analysis of the experience of SS through in-depth interviews has allowed for us to understand the particular meaning of this process and to document a whole typology of experiences of SS.

The categories and subcategories to which participants devoted most time in the interview were those related to avoidance and escape behaviours from their social environment, labels, loss of social relationships, negative repercussions and self-hiding of the problem, including the diagnosis. It was observed that the categories and subcategories revealed influence each other, forming a looping effect that ends up explaining and feeding the experience of SS. 

Participants revealed an unwanted relationship with a host of negative stereotypes and experienced that the people around them perceived them as dangerous, unpredictable and incapable. Qualitative and quantitative literature reviews have consistently identified these labels among the general public and across countries [8,9,18,19]. All these studies reported experiences of discrimination and rejection, the observation of fear in others when they are present as well as the experience of feeling contemptuously labelled just because they have a mental illness [19]. As in previous studies, our results also underline that the configuration of SS is not only influenced by external factors (neighbours, community, media and clinical staff) but also the most immediate and reference environment, such as the family, can contribute to its consolidation [18]. 

With regard to general social context factors, most participants reported experiencing discrimination in numerous social interactions and through media representations that reflect and reinforce these stereotypes. “Crazy”, “murderer”, “freak” and “moron” are some of the derogatory terms most often heard by people with psychosis. In addition, findings concerning factors in the immediate social environment, such as the family, should also be considered. In our study, participants experienced their families as both a source of support and a source of stigma. While many acknowledged their care, support and attention, they also perceived a certain lack of understanding of their illness, an unnecessarily overprotective treatment and often a tendency to silence or conceal when others were present. Corrigan et al. [32] had already mentioned that family shame prejudice, belief in responsibility for the diagnosis and/or in the possibility of parental transmission of the disease are factors contributing to the development of SS.

Many participants stated that discrimination in the workplace is also a consequence of negative stereotypes about the disease, particularly those related to incompetence. People in this study reported such stigmatisation at work and, as in the Ong et al. [19], revealed that this type of discrimination is reflected in aspects such as the obligation to report psychiatric history in job interviews. Alternatively, participants consider that their psychiatric condition should not determine their ability to work.

Regarding institutional stigma, some interviewees mentioned unequal treatment, barriers and a lack of humanization. This result is consistent with that observed in studies that showed how participants felt discriminated against during their treatment process or mentioned a lack of kindness and consideration for their views [18,19,33]. 

The results also reported a loss of social roles such as dropping out of school and/or loss of work. Other participants broke away from these roles at the onset of the illness and could no longer regain them due in part to SS. As we see, the internalisation of what we have called contextual stigma in this study can lead to the loss of previous social roles and the adoption of life-limiting avoidance behaviours that reinforce both contextual stigma and SS. This vicious cycle leads to feelings of inferiority and personal devaluation that are further exacerbated by recognising oneself as cognitively impaired and overwhelmed by the frequent interference of positive symptomatology. It is not surprising that all these experiences lead, time and again, to new and continuous avoidance. In fact, the loss of cognitive abilities was another category with which participants tried to explain the reasons for their isolation, the scarcity of their social relations and the inactivity that characterised their daily routines. These dynamics were also evident when participants linked their disability to their feelings of insecurity and the effects of the prescribed medication on their mood.

In addition to this, the categories and subcategories of the components of self-stigma theme were also associated with labels, derived emotional responses (prejudice), escape and/or avoidance behaviours and negative repercussions. This is consistent with what is found in the international literature on stigma in mental illness [17,18]. Participants reported internalising negative beliefs that they continually observed in their environment, alienation in the form of shame, fear of rejection and feelings of inferiority. This all leads to the self-isolation and self-discrimination characteristic of SS. Self-criticism and self-deprecation often result in the abandonment of work, treatment and help-seeking as well as the loss of opportunities for independent living. In this sense, as has been suggested in the previous literature, people with psychosis may benefit from intervention programmes that include strategies related to self-compassion, acceptance and cognitive defusion [25,34]. Lien et al. [35] found that guiding patients with a diagnosis of schizophrenia to perceive themselves more positively can effectively reduce their SS. Indeed, the tendency to avoid and try to get rid of negative self-evaluations has a negative impact on the social functioning of people with psychosis [25], it undermines their independence and feeds the experience of SS over and over again.

Concealment of one’s illness is another of the coping behaviours most commonly reported by participants. The previous literature has documented that both active avoidance and self-concealment represent paradoxical coping strategies that are commonly employed by people with psychosis [36,37,38]. Overall, everything seems to confirm that people with psychosis maintain a defensive and intolerant relationship towards their own disorder. Such self-concealment, carried out in principle to avoid stigma, paradoxically leads to further avoidance and isolation behaviours that circularly reinforce SS. 

The findings from listening to the first-person experiences of people with psychosis allow for us to conclude that the development of the process called SS is related to the incorporation of negative stereotypes of public stigma, experiences of discrimination and rejection and media representations of these images. 

Qualitative research provides an in-depth understanding of a phenomenon. Being able to access the experiences of our patients enables more sensitive, humanised and effective nursing interventions. It also reveals unseen aspects of care. The findings of this study can help us to transform our relationship with patients. Knowing their strengths and experiences is of great importance to enhance their recovery without reducing everything to their illness. It is also useful to guide the selection of relevant variables in future quantitative research or to delve into other emerging themes in a qualitative way.

### Limitations

This study is subject to the limitations of the methodology used. Nevertheless, all the criteria of rigour have been applied, guaranteeing a level of credibility, transferability, consistency and confirmability that is desirable at a theoretical level. The present study was performed with outpatients from a Mental Health Public Service in Asturias with specific sociodemographic and clinical characteristics. Therefore, the results are not generalisable but transferable to the type of context and sample. As our sample is limited, it is not representative of all individuals with chronic psychosis. The questions asked in the interviews required only subjective statements, thus there is no quantifiable measurement of the results.

As a strength, we consider the triangulation of methods carried out when selecting the participants with a high level of SS according to measures quantified in a previous study.

It would be interesting for future research to study the perception of people with psychosis with low SS scores to better understand their coping styles, and resilience. Triangulation of research data (family, mental health professionals) is also suggested for future studies of the SS phenomenon as it allows for a more holistic understanding of the evidence and may help identify problems or inconsistencies in the data.

## 5. Conclusions

This study explored the SS of patients with psychosis from the phenomenology approach proposed by Husserl. The results of the research revealed 4 themes and 15 categories. The observed themes were: contextual stigma, components of self-stigma, skill loss and coping with self-stigma. All these themes influence each other in a looping effect that explains and amplifies the experience of SS. Qualitative research is a useful tool to gain insight into the patient’s world and to intervene according to personally experienced needs. It is hoped that these findings will enable the implementation, in nursing practice, of strategies of cognitive defusion as well as acceptance and distancing aimed at minimising the impact that contextual stigma can have on people with psychosis. 

## Figures and Tables

**Table 1 ijerph-20-05688-t001:** Main socio-demographic characteristics. Frequencies (F) and percentages of the sample (%) (*N* = 14).

Socio-Demographic Characteristics	F	%
Sex		
Male	7	50
Female	7	50
Age		
Mean: 48.1		
SD: 10.2		
Marital status		
Single	7	50
Married/with partner	3	21.4
Separated/divorced	4	28.5
Total	14	100
Living arrangements		
Alone	3	21.42
Family of origin	10	71.42
With partner	1	7.14
Total	14	100
Education		
Primary	3	21.42
Secondary	10	71.42
University	1	7.14
Total	14	100
Employment situation		
Housework	1	7.14
Inactive (unemployed)	3	21.42
Active (employed)	1	7.14
Inactive (Employment disability)	8	57.14
Other	1	7.14
Total	14	100

**Table 2 ijerph-20-05688-t002:** Main clinical characteristics. Frequencies (F)/mean (M) and percentages (%) or standard deviation (SD) of the sample (*N* = 14).

Clinical Characteristics	F or M	% or SD
ICD-10 diagnosis		
Paranoid schizophrenia (F20.0)	9	64.28
Delusional disorder (F22)	2	14.28
Schizoaffective disorder (F25)	1	7.14
Schizotypal disorder (F21)	2	14.28
Total	14	100
Employment disability		
No	1	7.14
33–49%	6	42.85
50–64%	3	21.42
65 o >65	4	28.57
Total	14	100
Substance use		
No	9	64.28
Yes	4	28.57
Ex-user	1	7.14
Total	14	100
Age at onset of psychosis (years)	28.6	9.1
Diagnosis (years)	19.5	10.9
Pharmacological treatment (years)	17.6	9.8
Type of treatment		
Depot antipsychotic injections	10	71.4
Oral antipsychotic	4	28.5
Number of hospitalisations	2.6	1.8
ISMI		
Subscale: Alienation	14	100
Subscale: Stereotype endorsement	7	50
Subscale: Social withdrawal	11	78.57
Subscale: Perceived discrimination	11	78.57
Subscale: Stigma resistance	7	50
Self-stigma Total score	2	14.28

**Table 3 ijerph-20-05688-t003:** Interview questions.

Question
What is your day-to-day like? Would you like to do other things?
What does your mental health problem mean to you? How do you cope with it?
What do you think of yourself? Have there been any changes in your feelings about yourself or your body? Have you done something to change what you don’t like?
How is your relationship with your family and what feelings does it generate? How do you feel about your relationships with others?
Have you ever felt discriminated, can you give me an example? How do you think others perceive you?
What do you do in your free time? If your answer is negative, what do you need to change it?

**Table 4 ijerph-20-05688-t004:** Examples of meaning units, condensed meaning units and codes.

Meaning Unit	Condensed Meaning Unit	Code
You are not a number, you are not a folder, you are something else, you are a story, a person	You are a person	Lack of humanization
I think I’m not up to normal people…	Do not measure up	Inferiority
I am ashamed… that they see how I am and that they know that I have this disease	Shame for having the disease	Shame
There’s stuff on television about a schizophrenic who killed so many people, and… People know what I have, so they’re scared of me	Schizophrenic who killed so many people	Dangerousness

**Table 5 ijerph-20-05688-t005:** Results. Subcategories, categories and themes.

Subcategory	Category	Theme
Stigma in the media Workplace Stigma Institutional Stigma Family Stigma	Social stigma	
Own stereotypes Influence of family stigma	Social stigma per se	Contextual Stigma
Lack of opportunities Mistrust Lack of symptom control Lack of skills Financial problems Family support	Loss of social relationships	
Dropping out of studies Loss of job	Loss of social roles	
Negative stereotypes	Labels	
Alienation	Derived emotional responses	
Isolation	Avoidance and escape behaviours	Components of Self-Stigma
Fear of rejection	Anticipatory discrimination	
Negative affect Low self-esteem	Negative impacts	
Lack of concentration Conversational difficulties Difficulty in problem solving	Cognitive disturbances	Skills Loss
Lack of motivation	Abandonment of activities	
Religious practice Resignation about illness	Passive coping	
Seeking professional help Improving self-care	Active coping	Coping with Self-Stigma
Social distancing Self-concealment of the diagnosis	Avoidant coping	
Attempting self-harm	Ineffective coping	

## Data Availability

The data presented in this study are available on request from the corresponding author. The data are not publicly available due to privacy and ethical reasons.

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
