# Peer review of "Experiences of Self-Stigma in People with Chronic Psychosis: A Qualitative Study"

_ijerph, 2023, doi:10.3390/ijerph20095688_

Round 1

Reviewer 1 Report

The topic of the article is interesting from a clinical point of view. In general, the manuscript is well structured, the reliability of the results is beyond doubt. However, the manuscript needs correction of the style and syntax of English, and correction of some sections is also recommended.

Lines 35, 51, 53, etc.: Please use the abbreviation SS after you mentioned it for the first time.

Materials and Methods: Please explain the sample size and add inclusion/exclusion criteria. Add information about the pharmacotherapy of psychosis that the participants received at the time of the interview.

Results: The presented results are descriptive, but I recommend the authors to add figures or tables for clarity and to increase the interest of readers of this open access journal.

Discussion: I propose to put the Limitations in a separate section.

References: 13 references out of 34 older than 10 years (published earlier than 2012). Please update them. In accordance with this, make corrections to the sections of the Introduction and Discussion.

The manuscript needs minor edition of the style and syntax of English,

Author Response

We want to thank you for your time and suggestions. We have revised our manuscript following all your interesting comments.

Please see the attachment. All these changes have been marked in red in our manuscript.

We look forward to responding to your comments and again, we want to sincerely thank you for your interest in our manuscript.

Reviewer 2 Report

Thank you for the opportunity to review this article.

The study is very relevant and current.

Abstract: It should be restructured in order to respond to the following: introduction, objectives, materials and methods, results, discussion and conclusion. The authors address themes and then categories and subcategories. It is important to highlight the methodology for content analysis and use the respective concepts.

Introduction: the authors address the topic in general, which allows contextualizing the reader, however, they should present in this section the objective of the study and the research question.

Materials and methods: the phenomenological method is used, but they must specify which method was used and summarize what this method presupposes.

Was it a phenomenological study? But the authors defined themes in advance.

How did the authors ensure credibility, consistency, transferability, and confirmability in this study?

Results: There is some confusion between themes and categories, the authors must make explicit the phenomenological method used so that it is possible to understand the results.

Discussion: The authors use current and relevant bibliography to discuss the results. What are the main limitations of this study?

Conclusion: responds to the objective of the study

Author Response

(The authors gave the same response as above.)

Reviewer 3 Report

INTRODUCTION
- The introduction is rather short, perhaps some more information could be added related to the content covered (e.g., it would be useful to add some theoretical background, which - for readers less familiar with the field - would tell more about psychoses)?
- Did you set any research questions?

METHOD
- Tables 1 and 2 are a bit difficult to read, could you make them more clear?
- Ethical considerations: could you please perhaps consider moving these to the end of the text, in a dedicated section (Institutional Review Board Statement)?

RESULTS
- Lines 157-162: not necessary, as you have already presented this in the tables in the method.

DISCUSSION
- Could you perhaps expand a little on the section relating to the limitations of the study and ideas for further research?

Author Response

(The authors gave the same response as above.)

Round 2

Reviewer 1 Report

The authors have significantly improved the manuscript.

However, there are minor recommendations:

Please add the name of the first and second columns in Table 1 and Table 2.

A minor revision of the style of English is required.

Author Response

(The authors gave the same response as above.)

Reviewer 2 Report

Congratulations to the authors,

They responded adequately to the requested revisions.

Author Response

We want to thank you for your time and suggestions. Thank you sincerely for your interest in our manuscript.